# Siformer: Feature-isolated Transformer for Efficient Skeleton-based Sign Language Recognition

Muxin Pu
Monash University Malaysia
School of Information Technology
Subang Jaya, Selangor, Malaysia
muxin.pu@monash.edu

Mei Kuan Lim
Monash University Malaysia
School of Information Technology
Subang Jaya, Selangor, Malaysia
lim.meikuan@monash.edu

Chun Yong Chong
Monash University Malaysia
School of Information Technology
Subang Jaya, Selangor, Malaysia
chong.chunyong@monash.edu

## Abstract

Sign language recognition (SLR) refers to interpreting sign language glosses from given videos automatically. This research area presents a complex challenge in computer vision because of the rapid and intricate movements inherent in sign languages, which encompass hand gestures, body postures, and even facial expressions. Recently, skeleton-based action recognition has attracted increasing attention due to its ability to handle variations in subjects and backgrounds independently. However, current skeleton-based SLR methods exhibit three limitations: 1) they often neglect the importance of realistic hand poses, where most studies train SLR models on non-realistic skeletal representations; 2) they tend to assume complete data availability in both training or inference phases, and capture intricate relationships among different body parts collectively; 3) these methods treat all sign glosses uniformly, failing to account for differences in complexity levels regarding skeletal representations. To enhance the realism of hand skeletal representations, we present a kinematic hand pose rectification method for enforcing constraints. Mitigating the impact of missing data, we propose a feature-isolated mechanism to focus on capturing local spatial-temporal context. This method captures the context concurrently and independently from individual features, thus enhancing the robustness of the SLR model. Additionally, to adapt to varying complexity levels of sign glosses, we develop an input-adaptive inference approach to optimise computational efficiency and accuracy. Experimental results demonstrate the effectiveness of our approach, as evidenced by achieving a new state-of-the-art (SOTA) performance on WLASL100 and LSA64. For WLASL100, we achieve a top-1 accuracy of 86.50%, marking a relative improvement of 2.39% over the previous SOTA. For LSA64, we achieve a top-1 accuracy of 99.84%. The artefacts and code related to this study are made publicly online. [1]

## CCS Concepts

• **Computing methodologies** → **Supervised learning by classification**; **Interest point and salient region detections**; **Feature selection**; **Neural networks**; • **Human-centered computing** → **Human computer interaction (HCI)**.

## Keywords

Sign Language Recognition; Skeleton-based Approach; Human-centered Computing; Interpretability; Pose Rectification; Feature Isolation; Adaptive Inference; Transformers

**ACM Reference Format:**
Muxin Pu, Mei Kuan Lim, and Chun Yong Chong. 2024. Siformer: Feature-isolated Transformer for Efficient Skeleton-based Sign Language Recognition. In *Proceedings of the 32nd ACM International Conference on Multimedia (MM '24), October 28-November 1, 2024, Melbourne, VIC, Australia.* ACM, New York, NY, USA, 10 pages. https://doi.org/10.1145/3664647.3681578

## 1 Introduction

Sign languages (SL) [9] were invented to facilitate natural and intuitive communication, aiming to assist people with hearing or speech impairments for a better quality of life. Globally, there are around 466 million people with these disabilities and more than 300 SLs [1]. Mastering SL proficiency remains a challenge for the general public since it is deeply influenced by the corresponding spoken language[18, 41, 48] and the associated culture [28]. In light of significant advancements in machine learning, it is crucial to delve into SLR to facilitate smooth communication for those with hearing or speech impairments. In this work, we focus on isolated SLR, a key task in visual SL research, which aims to recognise SL at the word level and is a fine-grained classification problem.

Recently, skeleton-based methods have gained popularity in the SLR task [2, 23, 40] due to their computational efficiency and portability. These methods analyse dynamic patterns from sequences of body poses, represented by skeletal joints and facial landmarks extracted from pose estimators, to effectively recognise sign glosses, which are words associated with the signs. Pose estimation becomes crucial in this context to measure the position and movement of individual fingers, the hand, and other related features. However, these methods still encounter three major limitations in the realm of skeleton-based SLR:

Firstly, **non-realistic skeletal representation** is a key problem that is often overlooked in SLR research. The more realistic the features and patterns that the chosen pose estimator can extract, the deeper the understanding can be gained in the context of SLR. SOTA pose estimators suffer from dynamic circumstances or complicated backgrounds, where issues such as occlusion, illumination variations, and motion blur come into play from time to time [29], leading to a further decrease in the overall anatomical correctness, the precision and realism of the output skeletal representation.

[1] https://github.com/mpuu00001/Siformer.git

This problem serves as the root cause of the unstable performance of skeleton-based SLR models, which are trained on misleading information.

Secondly, **missing data** is another common problem in practice that has not been well explored in SLR research. Typical machine learning models assume complete data availability in both the training and inference phases. However, in reality, there is a high risk that certain data may be absent at either phase due to malfunctions during data collection, high data acquisition costs, human errors in data preprocessing, etc [45]. For instance, the joints of human hands are prone to occlusion or self-occlusion due to articulation or viewpoints. The hands, body parts, and faces may fail to be detected under the influence of motion blur and illumination variations [29]. As such, a mechanism that enhances SLR model robustness to missing data is essential.

Thirdly, **inefficient inference** is a problem generally caused by overthinking during the decision-making process. For many input samples, the shallow representations at an earlier layer are sufficient to make a correct classification, whereas the representations at the final layer may become distracted by over-complicated or irrelevant features that do not generalise well [53]. The complexity level of the sign glosses varies in SL. Some sign glosses need a mix of global body movements, delicate gestures performed by the highly articulated hand, and (micro-)facial expressions. In contrast, some sign glosses can be easily performed by a single hand. Existing SLR models typically handle all sign glosses similarly, regardless of their complexity level in terms of skeletal representations. This inefficiency may limit the deployment of SLR models in contexts where inference speed and computational costs are critical, as well as making them vulnerable to adversarial attacks.

To address the aforementioned limitation, this paper proposes the following solutions:

(1) We present a **kinematic hand pose rectification** approach that implemented constraints on the hand joints based on their kinematic constraints, since hand gestures play a dominant role in performing SL. These constraints derived from [11, 17, 31] define the allowable range of motion for each finger's flexion-extension and abduction-adduction movements. This approach enhances the realism of hand skeletal representations, resulting in more reasonable and lifelike representations of SL glosses by skeletal data.

(2) We propose a **feature-isolated mechanism** to handle hand gestures and body gestures separately, improving efficiency in attention calculation and easing the impact of detection failures of hands and other body parts. Based on our observations, detection failures of hands and other body parts are independent. They are usually estimated by different off-the-shelf estimators. The hand occupies a relatively smaller area, making the detection failure rates higher than other body parts. Commonly, Transformers rely heavily on the attention mechanism [42] to capture internal temporal or spatial relationships among features. Naturally, the movement of hand joints has less of a relationship with other body parts. Performing isolated encoding for hand gestures and body postures can enhance the efficiency of attention calculation and enable the capture of locally focused feature maps. This, in turn, reduces the impact of missing data on separate body parts.

(3) We design **input-adaptive inference** to dynamically adjust computational paths based on the complexity level of sign glosses. Drawing inspiration from [53], we implemented a patience-based early existing mechanism. This mechanism collaborates with inner classifiers for each layer and dynamically halts inference when the intermediate predictions of the internal classifiers remain unchanged for a predetermined number of consecutive times. This mechanism eases the impact of overthinking issues, thereby enhancing robustness and accuracy. We believe that in resource-limited scenarios (e.g., portable devices) or with larger datasets, it enables Siformer to efficiently reduce computational costs and achieve higher generalisation beyond the given datasets.

On the whole, a feature-isolated Transformer network, named Siformer, is proposed (depicted in Figure 3), which contains three main components: kinematic hand pose rectification (detailed in Section 3.2), feature-isolated mechanism (detailed in Section 3.3), and input-adaptive inference mechanism (detailed in Section 3.4).

## 2 Related work

### 2.1 Realistic skeletal representation

Previous research has partially explored the problem of pose realism, focusing on achieving highly accurate estimation to minimise position-based errors [38, 39, 49]. Despite low errors in benchmark tests, these studies often overlook the broader issue by solely prioritising estimator accuracy. In terms of error diagnosis in pose estimation, Ronchi and Perona [32] introduced a method to analyse errors in multi-instance pose estimation algorithms, along with a principled benchmark for comparison. Qualitative examples from their study reveal that a large portion of these errors tend to violate kinematic constraints and affect anatomical correctness, sparking increased research interest in the realism of poses. Issac et al. [17] proposed a single-shot corrective convolutional neural network to enforce kinematic constraint for depth-based hand pose estimation tasks. Zecha et al. [50] introduced a rectification pipeline to refine noisy joint coordinates estimated from swimmers via swap correction, outlier correction, and data-dependent joint refinement. However, their methods are designed for a specific modality or tailor-made and not plug-and-play. To the best of our knowledge, no existing work explores the impact of non-realistic skeletal representation and the effectiveness of rectification in SLR research.

### 2.2 Sign language recognition

Skeletal data has been used for efficient action recognition in many works [4, 7, 16, 24–26]. Architectures based on skeletal data have gained popularity in SLR tasks, with a focus on capturing both global body motion information and local arm, hand, or facial expressions simultaneously. Tunga et al. [40] proposed a skeleton-based SLR method that utilised GCN [20] to model spatial dependencies and BERT [6] for temporal dependencies among skeletal data in SLR. The two representations were eventually combined to determine the sign class. Boháček and Hrúz [2] presented SLR based on a Transformer model utilising skeletal data, featuring novel data augmentation methods and separate the space of hands and body based on the corresponding boundary values. Most approaches assume all necessary information for SL recognition can be obtained from the pose of the signer's pose, hands, and, optionally, face. However, in reality, certain features may not be adequately captured. Handling missing data is a common challenge, and many

techniques exist for handling it. However, given the complexity of human motion and the nuanced flexibility of hands, simple deletion or imputation may be too casual. Consequently, to mitigate the impact of this problem, we present a novel method.

### 2.3 Efficient inference

Research aimed at enhancing the efficiency of deep neural networks can be broadly classified into two categories: 1) Static approaches [10, 27, 36, 37, 43, 47] involve designing either compact models or the compression of heavy models. These models maintain a fixed structure for all instances during inference, implying that the input consistently traverses the same layers. 2) Dynamic approaches [14, 19, 33, 44, 46, 53] permit the model to select varying computational paths based on each instance during inference. This flexibility in computational paths allows for adaptability and customisation. In alignment with the inspiration drawn from these existing works, Siformer is strategically designed using a dynamic approach for inference. The dynamic nature of our model allows it to adjust its computational paths based on the characteristics of individual sign glosses during the inference process.

## 3 Methodology

### 3.1 Preprocessing

To mitigate potential issues arising from class distribution imbalance, one of the simplest strategies for achieving balance is to randomly remove majority class samples from the training set or replicate minority class samples [8]. However, considering the scarcity of data in the domain of SLR, the straightforward removal of samples is not suitable for our case. We opt to employ the synthetic minority over-sampling technique (SMOTE) to generate synthetic data samples, thereby achieving a balanced training set. SMOTE stands out as the most well-known oversampling method [8]. It uses the K-nearest neighbours algorithm to identify neighbouring minority class samples and generate new samples by incorporating these identified neighbours.

**SMOTE**. Before employing SMOTE, we standardise the frame numbers for each sample. This standardisation involves aligning the frame numbers based on the maximum number of frames by padding additional zeros at the end of the corresponding skeletal data. The padded zeros have minimal to no impact on computational efficiency, because of the nature of the attention mechanism (detailed in Section 3.3) we adopted. Subsequently, we identify the class with the maximum number of samples in the training set, denoting this maximum number as $N$. The process of SMOTE can be expressed as follows:

$$x_i' = x_i + w(x_k - x_i) \tag{1}$$

For each class, $x_i$ denotes a randomly selected minority sample, and the K-nearest neighbours of $x_i$ are identified among the minority samples. From these neighbours, $x_k$ is randomly selected. Then, $x_k$ and $x_i$ are utilised in conjunction to conduct linear interpolation, yielding the new synthetic sample $x_i'$. Here, $w$ represents a uniform random variable within the range of $[0, 1]$. This process iterates until each class contains $N$ samples.

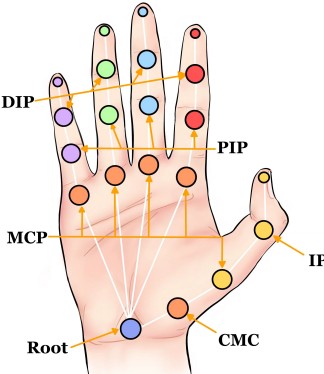

**Figure 1: The structure of the hand skeleton along with the respective joint names, including distal interphalangeal (DIP), proximal interphalangeal (PIP), metacarpophalangeal (MCP), carpometacarpal (CMC), interphalangeal (IP) joints, based on [11, 17, 31].**

### 3.2 Kinematic hand pose rectification

The realism of skeletal presentations is often overlooked in SLR research. Existing SLR models are usually trained on non-realistic skeletal data, which could mislead the recognition and yield low accuracy. To address this challenge, we develop a rectification procedure in Siformer aimed at adjusting the individual joint angles of hand poses. The abduction, adduction, extension, and flexion of the fingers, as illustrated in Figure 2, are well-explored in anatomy; they represent crucial kinematic information that differentiate sign

**Table 1: Kinematic constraints for the allowable range of motion for each finger's joint derived from [11, 17, 31].**

| Motion | Joint | Min (∘) | Max (∘) |
|---|---|---|---|
| Abduction and Adduction | Thumb CMC | 0 | 45 |
| | Thumb MCP | -7 | 12 |
| | Other Finger MCP | -15 | 15 |
| Extension and Flexion | Thumb CMC | -20 | 45 |
| | Thumb MCP | 0 | 80 |
| | Thumb IP | -30 | 90 |
| | Other Finger MCP | -40 | 90 |
| | Other Finger PIP | 0 | 130 |
| | Other Finger DIP | -30 | 90 |

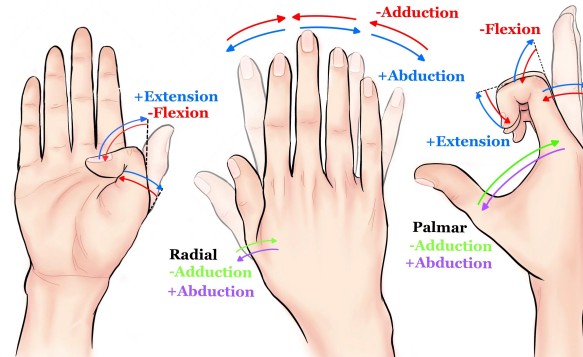

**Figure 2: Examples of adduction, abduction, flexion, and extension of hands are referenced in [3, 11, 31].**

glosses. The active ranges of motion for each finger's joint are summarised in Table 1, and the hand structure along with corresponding joint names is illustrated in Figure 1. By reducing both lateral and angular deviations in movement based on kinematic information derived from empirical data [11, 17, 31], our rectification process yields refined skeletal data that more accurately reflect intended gestures and movements, distinguish similar gestures by providing detailed data on the dynamics of movement, and ultimately bolstering the performance in recognising SL gestures.

**Rectification**. The rectification procedure (depicted as (1) in Figure 3) leverages kinematic constraints to adjust the hand to the nearest possible pose if the constraints are violated. The following are the equations utilised for the implementation:

$$\theta_i = \cos^{-1}\left(\frac{P_i(t) \cdot P_r(t)}{||P_i(t)||_2 ||P_r(t)||_2}\right) \tag{2}$$

$$\varepsilon_i^\theta = f(\theta_i) = \begin{cases} \theta_i - \theta_{max} & \text{if } \theta_i > \theta_{max} \\ \theta_{min} - \theta_i & \text{if } \theta_i < \theta_{min} \\ 0 & \text{otherwise} \end{cases} \tag{3}$$

$$P_i(t)' = R(\alpha \varepsilon_i^\theta) \cdot P_i(t), \text{ if } \varepsilon_i^\theta > 0 \tag{4}$$

In the procedure, we first compute the joint angle $\theta_i$ using Equation 2, where the temporal location of predicted joint $P_i(t)$ describes the position of joint $i$ at time $t$, and $P_r(t)$ represents the reference joint of the corresponding joint $i$. The joint angle $\theta_i$ is then subject to correction as follows: if $\theta_i$ is larger than the defined maximum angle $\theta_{max}$, the error value $\varepsilon_i^\theta$ is calculated using Equation 3 by $\theta_i$ minus $\theta_{max}$. If $\theta_i$ is smaller than the defined minimum angle $\theta_{min}$, $\varepsilon_i^\theta$ is calculated using Equation 3 by $\theta_{min}$ minus $\theta_i$. If $\varepsilon_i^\theta = 0$, no rectification is applied; otherwise, $P_i(t)$ will be rotated based on $\varepsilon_i^\theta$ using Equation 4 with the rotation matrix $R$. The actual rotation direction, whether clockwise or counterclockwise, is determined by the direction of $P_r(t)$ towards $P_i(t)$. Since the procedure rectifies hand poses based on the kinematic constraints, it is inevitable that the hand pose drifts from its initial position. This drift has the dual potential to either unveil pivotal revelations that lead to precise classification or to introduce trivial details that manifest as noise or disturbance within the classification process. Therefore, the alpha value $\alpha$ is designed to control the rectification. When $\alpha = 0.2$, the angle is 20% rectified based on the hand's kinematic constraints. When $\alpha = 1$, the angle undergoes complete rectification, adhering to the hand's kinematic constraints by 100%.

### 3.3 Feature-isolated Transformer

In contrast to existing methods that aim to capture intricate relationships among different body parts collectively, we introduce a novel architecture (depicted in Figure 3) for Siformer. This architecture allows us to focus on capturing local temporal-spatial context concurrently and independently from individual features during the encoding phase. We can then tailor settings for different body parts. Following this, we concatenate the feature maps for decoding. This approach effectively decouples the strong interdependence between features and eases the impact of missing data on individual body parts during the recognition process. Additionally, we design learnable frame-wise positional encoding for Siformer to incorporate positional information into the sequence and leverage

the ProbSapre self-attention mechanism to streamline space and time computation efficiently.

**Feature-isolated mechanism**. Given the dominant role of the hands in SL, we decoupled the hand gestures and body gestures during the encoding process to capture locally focused feature maps from individual features (depicted as (2) in Figure 3). These feature maps are then concatenated as:

$$F_{map} = concat(F_l, F_r, F_b) \in \mathbb{R}^{(L_l + L_r + L_u) \times d} \tag{5}$$

where the feature maps obtained from the left hand gestures $F_l$, right hand gestures $F_r$, and upper body posture $F_b$ are concatenated as $F_{map}$ to be fed into the decoder. We employ the class query-based decoder [2], where only one class query is needed to represent the targeted sign gloss. The concatenated feature map $F_{map}$ and the projected class query are processed by the multi-head attention projection module, followed by a linear layer with a number of neurons equal to the number of classes, and the softmax activation is used to predict the confidences of each class.

**Learnable frame-wised positional encoding**. The self-attention operation in Transformers is permutation-invariant, meaning it disregards the order of tokens in an input sequence. To reintroduce the importance of order, positional encoding becomes crucial.

In this work, we design learnable frame-wise positional encoding. The initial encoding can be represented as follows:

$$X' = X + P \tag{6}$$

where $P_{i,i} = P_{i,j}, P_{i,i} \neq P_{j,i}$ and $i \neq j$. This means that each skeletal data within a frame shares an identical positional value, while skeletal data corresponding to the same keypoint across different frames are assigned different positional values. This approach ensures that positional values vary across frames but remain consistent within a frame. The input $X$ and the learnable positional embedding matrix $P$ share the same dimension. $P$ initially contains random values, and it is jointly updated as one of the network parameters during the training process.

**ProbSpare self-attention mechanism**. Originally, the Vanilla self-attention mechanism [42] takes each tuple input from the packed matrices $Q \in \mathbb{R}^{L_Q \times d}$, $K \in \mathbb{R}^{L_K \times d}$ and $V \in \mathbb{R}^{L_V \times d}$ for queries, keys and values respectively, to perform scaled dot-product, which can be expressed as:

$$A(Q, K, V) = softmax\left(\frac{QK^T}{\sqrt{d}}\right)V \tag{7}$$

This approach of performing scaled dot-product requires quadratic time computation and $O(L^2)$ space computation, posing a major drawback when enhancing prediction capacity. Previous attempts have shown that the distribution of Vanilla self-attention probabilities exhibits potential sparsity. Researchers have proposed "selective" counting strategies to ease the impact of this drawback on time or space efficiency without significantly compromising performance.

Based on this insight, we analysed the distribution of Vanilla self-attention scores in the context of skeleton-based SLR. Upon reviewing the heat maps of attention scores, we noticed a trend where certain dot-product pairs made major contributions to the primary attention, while others yielded negligible attention. To enhance attention efficiency, we adopted the ProbSpare self-attention mechanism [52]. This method improves feature retrieval by focusing

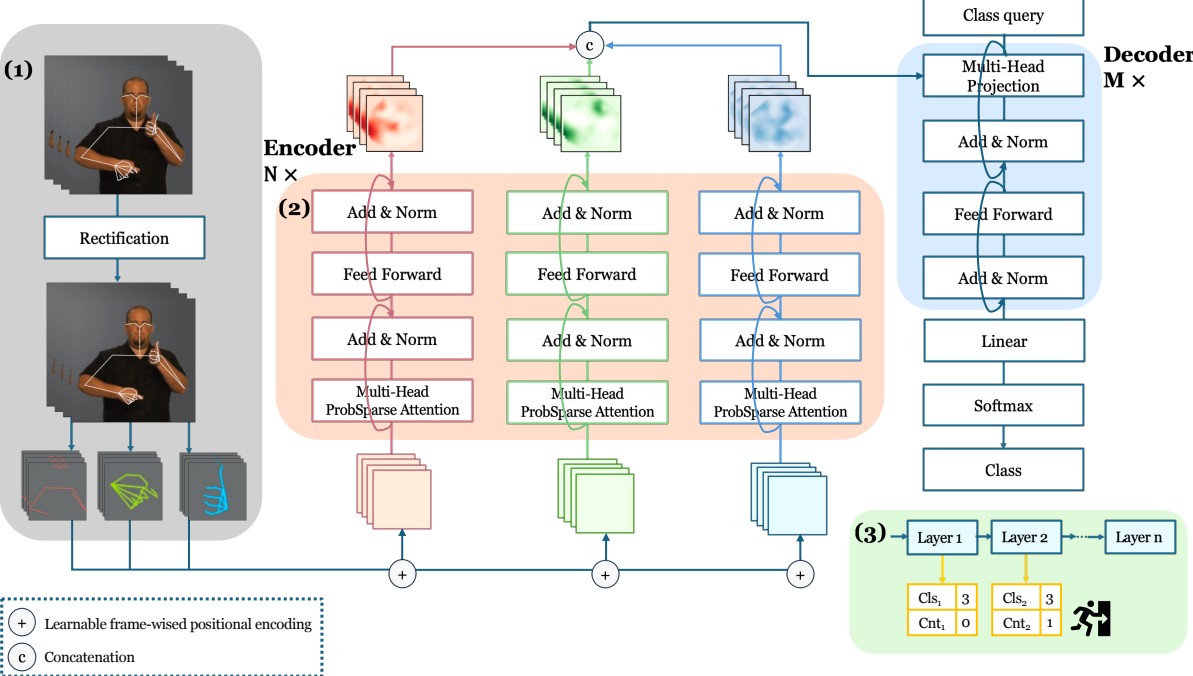

**Figure 3: The core components of our proposed method Siformer: (1) Kinematic rectification is applied to correct poses of sign glosses, aiming to provide realistic representations. (2) We propose a feature-isolated mechanism that captures local spatial-temporal context concurrently and independently from individual features during the encoding phase. This is followed by combinatorial-dependent decoding. (3) We integrate an internal classifier at each layer to achieve input-adaptive inference. The provided example illustrates a case when the patience value is set to 1.**

on the most relevant feature pairs. It identifies important features more effectively than Vanilla self-attention by allowing each key to selectively attend to the top $u$ dominant queries. This approach focuses attention on the most relevant information, thereby reducing unnecessary computational overhead. The ProbSpare self-attention mechanism is expressed using the following equation:

$$A(Q, K, V) = softmax(\frac{\overline{Q}K^T}{\sqrt{d}})V \qquad (8)$$

where $\overline{Q}$ is a sparse matrix of the same size as $Q$, containing only the top-$u$ queries under the sparsity measurement:

$$M(q_i, K) = \max_j\{\frac{q_i k_j^T}{\sqrt{d}}\} - \frac{1}{L_k}\sum_{j=1}^{L_k}\frac{q_i k_j^T}{\sqrt{d}} \qquad (9)$$

As highlighted in [52], the calculation of $M(q_i, K)$ requires only $L_K \ln L_Q$ random samples to form dot-product pairs. The remaining pairs are strategically filled with zeros to enhance efficiency and numerical stability. The top-$u$ queries are then selected, to form $\overline{Q}$. In practical scenarios, the input lengths of queries and keys typically align in self-attention computation, denoted as $L_Q = L_K$. Consequently, the total time complexity and space complexity of ProbSparse self-attention amount to $O(L \ln L)$, which is more efficient than the Vanilla full self-attention mechanism, especially for long SL sequences. We experimented with a pyramid-structured encoder to distil self-attention feature maps. However, instead of

improving the performance of Siformer, the distillation process resulted in the model becoming heavier and less efficient.

## 3.4 Input-adaptive inference

For the sake of efficient inference and automatic adjustment, we integrate an internal classifier $Cls$ for each layer in our proposed Siformer, drawing inspiration from [53]. The mechanism of input-adaptive inference is visually represented as (3) in Figure 3. The patience counter $cnt$ is introduced to track the number of consecutive occurrences where predictions remain "unchanged", and the counter is defined as:

$$cnt_i = \begin{cases} cnt_{i-1} + 1 & \text{if } y_i = y_{i-1} \\ 0 & \text{if } y_i \neq y_{i-1} \lor i = 1 \end{cases} \qquad (10)$$

$$y_i = arg\,max(Cls_i(h_i)) \qquad (11)$$

where $y_i$ is a layer prediction outputted by the $i$-th internal classifier $Cls_i$ on the hidden state $h_i$ at $i$-th layer, The $arg\,max$ function returns the index of the maximum probability from the input probability vector. In contrast to [53], these internal classifiers are excluded during the training process. We experimented with both trained internal classifiers and "brand-new" internal classifiers. We found that the two methods produced similar results, with "brand-new" internal classifiers offering an improvement of 1.03% over the trained classifiers.

## 4 Experiments

### 4.1 Implementation details

**Datasets.** We conduct experiments for our proposed Siformer on two benchmark datasets: the WLASL100 [23], which is a word-level subset of American SL, and the LSA64 dataset [30], which holds Argentinian sign language. The two datasets were chosen for their more complete annotations and variation in terms of demographic diversity, making them more suitable for our experiment. The WLASL100 dataset consists of 2,038 videos featuring 100 distinct sign glosses performed by native American SL interpreters or signers. This dataset was collected from various public resources primarily designed for SL learning. Each sign gloss was performed by a diverse group of people of different races, and each gloss class comprises a varying number of samples. The LSA64 dataset contains 3200 videos featuring 64 distinct sign glosses from Argentinian SL. These glosses were carefully selected from the most frequently used terms in the LSA lexicon, covering both verbs and nouns. The videos were performed by 10 non-expert participants, each repeating each sign gloss 5 times. The skeletal representations of both datasets are estimated by Boh'aček and Hr'uz [2]. Among the 64 gloss classes in the LSA64 dataset, only 7 classes contain fewer samples than the maximum. The number of samples in the LSA64 dataset is relatively balanced across classes. Therefore, we only oversampled data from the WLASL100 dataset and left the number of samples in the LSA64 dataset unchanged. The LSA64 dataset is used with a random split of 80% for the training set and the remaining 20 % for the testing set. Before oversampling data in the WLASL100 dataset, we randomly sampled 8 samples from each of the 100 classes for the testing set. The oversampling process, as described in section 3.1, only took place on the remaining 2400 samples. Consequently, we have 3200 samples for the WLASL100 training set and 800 samples for the WLASL100 testing set.

**Training details.** We conduct training on both datasets over 100 epochs using an AdamW optimiser with $\beta_1 = 0.9$ and $\beta_2 = 0.999$, coupled with a weight decay rate of $1 \times 10^{-8}$. A MultiStepLR scheduler is applied with a decay factor of $\gamma = 0.1$. Initially, the learning rate is set to $1 \times 10^{-4}$, subsequently decaying at the 60th and 80th epochs. For the loss function, we utilised the standard cross-entropy loss, and the weights are randomly initialised.

### 4.2 Rectification analysis

The alpha $\alpha$ value in Equation 4 of Section 3.2 governs the extent to which the hand undergoes rectification based on kinematic constraints. We investigate different $\alpha$ values, including 0, 0.2, 0.4, 0.6, 0.8, and 1, where $\alpha = 0$ indicates no rectification applied. We find that rectification remains effective regardless of the specific $\alpha$ value utilised. Comparison of the radar areas for abduction-adduction (AA) rectification and flexion-extension (FE) rectification, as depicted in Figure 4, reveals a consistent trend: the effectiveness of both AA rectification and FE rectification exhibit a decrease from $\alpha = 0.2$ to $\alpha = 0.4$, followed by a moderate increase from $\alpha = 0.4$ to $\alpha = 0.6$, and then a continuous decline until $\alpha = 1$. The distances between the data points along each axis signify the disparity. In contrast to the trends observed in the effectiveness of individual rectification (AA and FE), optimal performance of combinational rectification is achieved when $\alpha = 0.4$, which reflects the largest

disparity between the effectiveness of individual rectification and the combinational rectification. This suggests that the two types of kinematic rectification do not have a cumulative effect on characterising the skeletal representations of different sign glosses. Instead, when striving for skeletal representations closer to reality, capturing trivial details becomes imperative for achieving high SLR accuracy. Our analysis shows that an $\alpha$ value of 0.4 provided the best trade-off between noise reduction and keypoints integrity, enhancing SLR performance without over-smoothing keypoints.

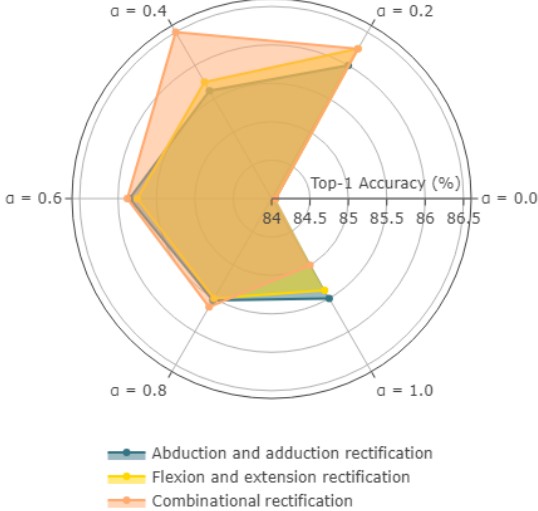

**Figure 4: Rectification analysis based on the variations of alpha $\alpha$ values on the WLASL100 datatset**

### 4.3 Effectiveness of input-adaptive inference

Before experimenting with our input-adaptive inference mechanism, we explored the effects of different numbers of encoder and decoder layers independently, while keeping other configurations consistent. As listed in Table 2, we can observe changes in accuracy corresponding to the variations in the number of layers. The highest accuracy is attained with 3 encoder layers and 2 decoder layers (85.38%), followed by the second-best performance with 3 encoder layers and 5 decoder layers (84.13%). Unless otherwise specified, we proceed to investigate the impact of the patience value using

**Table 2: Performance analysis without input-adaptive inference on the WLASL100 datatset**

| Encoder layer | Decoder layer | Top-1 Accuracy (%) |
|---|---|---|
| Variations in the number of encoder layers | | |
| 2 | 2 | 84.38 |
| **3** | **2** | **85.38 (C1)** |
| 4 | 2 | 83.63 |
| 5 | 2 | 80.50 |
| 6 | 2 | 76.88 |
| Variations in the number of decoder layers | | |
| 3 | 2 | 85.38 |
| 3 | 3 | 84.00 |
| 3 | 4 | 84.10 |
| **3** | **5** | **84.13 (C2)** |
| 3 | 6 | 83.88 |

**Table 3: Performance analysis with input-adaptive inference on the WLASL100 datatset**

| Patience value | Top-1 Accuracy (%) | Avg. Inference time (seconds) | Avg. FLOPs (G) | Params (M) |
|---|---|---|---|---|
| C1 with input-adaptive inference based on the encoders | | | | |
| **1** | **86.50** | **0.0324** | **0.46** | **2.55** |
| 2 | 85.73 | 0.0332 | 0.62 | 2.55 |
| C2 with input-adaptive inference based on the decoder | | | | |
| 1 | 85.75 | 0.0363 | 0.60 | 4.18 |
| 2 | 86.00 | 0.0345 | 0.61 | 4.18 |
| 3 | 86.00 | 0.0347 | 0.62 | 4.18 |
| 4 | 86.00 | 0.0345 | 0.64 | 4.18 |

**Table 4: Quantifying the effectiveness of input-adaptive inference with a patience value of 1 and C1 configuration on the WLASL100.**

| Total samples count | Early exit cases count |
|---|---|
| **800** | 41 |
| 2400 | 131 |
| 4000 | 195 |

the two best-performing layer configurations, which are bolded in Table 2. This analysis aims to leverage the optimal encoder and decoder layer configurations to determine the optimal patience value for our input-adaptive inference mechanism. Unlike training, inference operates on a per-instance basis, which ensures adaptive inference for processing individual data. This practice is also commonly employed in latency-sensitive production scenarios [33].

As illustrated in Table 3, different patience values can lead to differences in both speed and performance. For configurations with 3 encoder layers and 2 decoder layers (C1), where the internal classifiers are linked with the encoders, the best balance between accuracy and speed is observed with a patience value of 1. This implies that our input-adaptive mechanism allows Siformer to terminate at a very early stage of inference. Earlier layers of the model adeptly capture visual features for classifying many skeletal representations of SL glosses, while deeper layers risk over-complicating the process with irrelevant or overly complex features, thus reducing generalisability. We qualified the number of early exit cases with a patience value of 1 and C1 configuration in Table 4 (we provide a detailed analysis in the Appendix). The bolded values indicate the number of samples from testing sets. By comparing Table 3 and Table 4, we observed that 5% of early exit testing cases result in an approximately 26% reduction in computational complexity, decreasing from 620 million to 460 million floating-point operations (FLOPs). As patience values increase, accuracy tends to decrease. Notably, a significant increase in FLOPs is observed when the patience value for C1 rises from 1 to 2, representing the largest increase across all settings. Conversely, performance sees slight enhancements with higher patience values in configurations with 3 encoder layers and 5 decoder layers (C2), where the internal classifiers are linked with the decoder. Similarly to C1, longer patience for C2 also results in a corresponding increase in FLOPs. Interestingly, there is minimal impact on inference time across all patience settings, and parameters remain constant regardless of patience due to the exclusion of internal classifiers from training. Based on our analysis, we believe that in practical applications

where data volume and variability are higher, the percentage of successful early exit cases would also increase.

## 4.4 Cope with missing data

The architecture of Siformer is designed to capture the unique characteristics of hand and body gestures independently and concurrently during the encoding phase and subsequently integrate these captured characteristics cohesively during the decoding phase. By isolating features in this manner, our approach ensures the thorough capture of individual features and easing the potential influence of missing data from other features. To evaluate our model's robustness against missing data, we conducted intensive experiments on testing sets featuring varying degrees of occlusion by removing data from different body parts. This process is motivated by the emulation of real-world scenarios encountered in capturing SL, where occlusion may occur in certain frames, resulting in the loss of some joints or keypoint data. The hand's skeletal representation comprises 21 keypoints (refer to Figure 1), and the upper body consists of 12 keypoints, including the nose, neck, two ears, eyes, shoulders, elbows, and wrists. Therefore, we prepared 3 testing sets to evaluate the robustness of Siformer against varying levels of occlusion on the left hand, right hand, and upper body, respectively.

As illustrated in Figure 5, the performance exhibits acceptable fluctuations, hovering between approximately 79.10% and 77.80% across all testing sets, subsequent to a decline from 86.50%. The peak accuracy of 86.50% was achieved under the configuration of

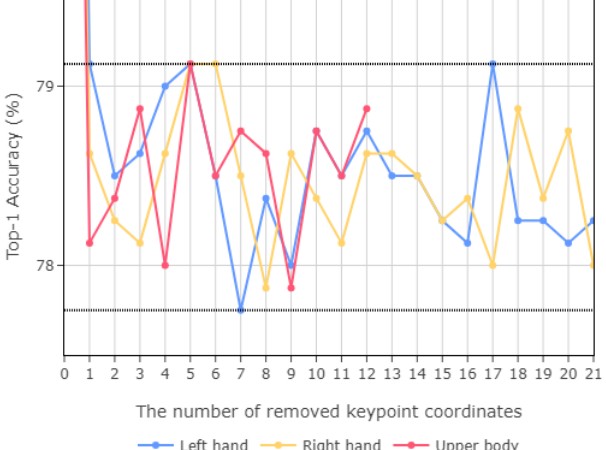

**Figure 5: Robustness testing against missing data on the WLASL100 datatset**

3 encoders, 2 decoders, and a patience value set to 1, with no removal of keypoint coordinates from either of the testing sets. The removal of a single keypoint coordinate from either testing set leads to a decrease in performance. Particularly, the removal of a keypoint from the upper body incurs a slightly greater degradation (by 7.48%) compared to other sets. This is followed by a decrease of 6.98% when one keypoint is removed from the right hand, and a decrease of 6.48% when one keypoint is removed from the left hand. It is worth noting that the performance degradation of our model is not continuous to the point of reaching its lowest, but rather it decreases and then improves after dropping. For example, removing five keypoints from either of the sets leads to a resurgence in the model's performance; in such a case, the accuracy scores obtained even exceed or remain the same as that of removing one keypoint from either of the sets separately. Overall, the largest fluctuation amplitude of the model's performance takes place between the number of removed keypoints of 1 to 12. The amplitude decreases after the number of removed keypoints reaches 12, and then increases after the number of removed keypoints reaches 16. We can now claim that our model is robust towards missing data because even when one of the body parts is entirely missing, the model's accuracy can still reach over 78.00%. This is observed when the number of removed keypoints equals 12 or 21.

## 4.5 Contribution of the core components

The results from our ablation study investigating the effects of each component in the proposed method are presented in Table 5. To evaluate the importance of each component in contributing to the overall performance of Siformer, we first trained Siformer using the optimal configuration. This optimal configuration includes 3 encoders and 2 decoders, with a patience value set to 1 for inference and an alpha value of 0.4 for both rectification methods. Additionally, there are 3 attention heads employed by both the left-hand and right-hand encoders, 2 attention heads designated for the body encoder, and 6 attention heads utilised in the decoder. As shown in Table 5, removing kinematic hand pose rectification results in a drop in top-1 accuracy by 2.45% from 86.50% to 84.05%; replacing our feature-isolated Transformer by Vanilla Transformer leads to a further decrease in accuracy by 6.98 % from 84.05% to 77.07%; omitting input-adaptive inference further decreases accuracy by 1.13% from 77.07% to 76.75%.

**Table 5: Ablation study on WLASL100 dataset for Siformer**

| Variations | Top-1 Accuracy (%) |
|---|---|
| Siformer | 86.50 |
| (-) Kinematic hand pose rectification | 84.05 |
| (-) Feature-isolated Transformer | 77.07 |
| (-) Input-adaptive inference | 75.94 |

## 5 Comparison with SOTA methods

In our comparison with previous SOTA methods on the two benchmark datasets, we organise methods based on their input modalities, specifically RGB-based methods and skeleton-based methods. As presented in Table 6, the SOTA methods generally achieve high performance on LSA64, and we boost the performance slightly. When comparing our method on WLASL100, which contains fewer samples compared to LSA64, it is notable that Siformer, utilising

**Table 6: Performance comparison on the WLASL100 and LSA64 datasets. Scores in parentheses are reported by the original authors, while the scores before the parentheses are from our reproduction using their GitHub source code.**

| Dataset | Method | Top-1 Accuracy (%) |
|---|---|---|
| | **RGB-based** | |
| | I3D [15] | 65.89 |
| | TCK [5] | 77.52 |
| | SignBERT+ [13] | 84.11 |
| | Fusion-3 [12] | 75.67 |
| WLASL100 | **Skeleton-based** | |
| | Pose-TGCN [15] | 55.43 |
| | ST-GCN [35] | 50.78 |
| | SignBERT+ [13] | 79.84 |
| | SPOTER [2] | 58.52 (63.18) |
| | Siformer (Ours) | **86.50** |
| | **RGB-based** | |
| LSA64 | LSTM + LDS [22] | 98.09 |
| | DeepSign CNN [34] | 96.00 |
| | MEMP [51] | 99.06 |
| | I3D [15] | 98.91 |
| | **Skeleton-based** | |
| | SPOTER [2] | 99.52 (100.00) |
| | LSTM + DSC [21] | 92.15 |
| | Siformer (Ours) | **99.84** |

only skeletal features as the input modality, even surpasses the performance of the most challenging RGB-based method [13] by 2.39%. This demonstrates the effectiveness of our proposed Siformer in leveraging skeletal features for SLR, showing its advantages even in datasets with limited sample sizes.

## 6 Conclusion

We introduce a novel feature-isolated Transformer, named Siformer, that achieves new SOTA performance across all two benchmark datasets. Unlike previous methods that neglect realistic hand poses, overlook the independence of body parts, and treat sign glosses uniformly regardless of complexity, we proposed novel solutions to overcome these limitations. Firstly, we enhance hand gesture realism through kinematic pose rectification. Secondly, we introduced a feature-isolated mechanism to capture local spatial-temporal context independently and concurrently, mitigating the impact of missing data and enhancing the robustness of the models. Thirdly, we presented an input-adaptive inference approach to adapt varying complexity levels of sign glosses, optimising both computational efficiency and accuracy. Experimental results validate the effectiveness of each novel component on overall performance. **Limitation:** Subtle changes in facial expressions are crucial for distinguishing confused sign glosses; we hope to explore their incorporation in future work. **Broader impact:** The lightweight nature of Siformer facilitates practical implementation on portable devices, benefiting online SL learning, daily communication, and SL typing methods. We hope our research contributes to practical application in the field of SLR and opens avenues for future exploration in enhancing the accessibility and efficiency of SLR.

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
