# OpenReview forum: "Siformer: Feature-isolated Transformer for Efficient Skeleton-based Sign Language Recognition"
_acmmm.org/ACMMM/2024/Conference — MM2024 Poster_

### Official Review · Reviewer_jQAk · 2024-05-15

**Rating:** 3
**Confidence:** 3

**Summary:**

This paper proposes a skeleton-based method for sign language recognition. The proposed method involves a data pre-processing strategy (kinematic hand pose rectification), a multi-stream feature fusion network (Feature-isolated Transformer), and an adaptive inference strategy to reduce computational costs. Experimental results demonstrate the effectiveness of these designs.

**Strengths:**

1) Skeleton-based sign language recognition is a promising task with potential applications and significant impacts on social welfare.
2) The proposed methods, with three key designs, are clearly introduced and easily implementable.
3) The ablation studies are comprehensive.

**Limitations:**

1) My major concern is about the novelty of the proposed method. It seems that the three contributions in this paper are all derived from existing work. The authors did not make significant technological modifications to them. Specifically, the kinematic hand pose rectification is a simple data pre-processing method that filters noise from the predicted hand keypoints. However, the proposed model doesn't exploit any kinematic information about the skeleton of the hand and the human body. The Transformer architecture and the adaptive inference strategy are also widely leveraged in many related fields.
2) More recent RGB-based and Skeleton-based action recognition methods should be compared in Table 5. The author may miss some SOTA methods in the experiment, e.g.[HD-GCN ICCV2023].
3) What do the self-attention maps illustrated in Figure 4 signify? Why is ProbSparse attention considered superior to Vanilla self-attention? Is the attention of the left hand focused on the most relevant body parts?
4) The author utilized two figures (Figure 1 and Figure 2) to illustrate the structure and examples of hand poses. However, this information might not be crucial to the main contribution. These figures should be simplified.
5) From Table 3, it appears that the reduction in computational complexity resulting from input-adaptive inference is not significant. What is the situation in practical application?

**Suitability:**

2

---

### Official Review · Reviewer_bkRe · 2024-05-19

**Rating:** 3
**Confidence:** 2

**Summary:**

The paper introduces Siformer, a Feature-Isolated Transformer designed for efficient skeleton-based Sign Language Recognition (SLR). Siformer addresses three primary limitations of current SLR methods: the neglect of realistic hand poses, the assumption of complete data availability, and the uniform treatment of all sign glosses regardless of complexity. To overcome these challenges, the authors propose a kinematic hand pose rectification method, a feature-isolated mechanism for local spatial-temporal context capture, and an input-adaptive inference approach. Experimental results demonstrate that Siformer achieves state-of-the-art performance on the WLASL100 and LSA64 datasets。

**Strengths:**

1. The paper introduces novel solutions to address significant limitations in current SLR methods, such as realistic hand pose rectification and a feature-isolated mechanism, enhancing the model's robustness and accuracy.
2. Very clear and impressive visualization and visual description in Figures 1, 2, and 3.
3. The paper presents extensive experimental results on two benchmark datasets, WLASL100 and LSA64, demonstrating the effectiveness of Siformer and achieving state-of-the-art performance.

**Limitations:**

1. While the proposed methods are somehow novel, they primarily build upon existing Transformer architectures and mechanisms. This might be seen as incremental improvements rather than groundbreaking innovations in the field of SLR. Please provide more explanation of the overall structure.
2. More datasets and evaluations are advised.
3. The explanation of the attention map visualization in Figure 4 is not clear.

**Suitability:**

3

---

### Official Review · Reviewer_Spwj · 2024-05-25

**Rating:** 4
**Confidence:** 2

**Summary:**

The paper introduces Siformer, a feature-isolated transformer architecture for efficient skeleton-based sign language recognition. The authors propose three key components: kinematic hand pose rectification, feature-isolated mechanism, and input-adaptive inference. Kinematic hand pose rectification improves the realism of hand skeletal representations by enforcing constraints derived from empirical data. The feature-isolated mechanism captures local spatial-temporal context independently for hand gestures and body gestures, enhancing robustness against missing data. Input-adaptive inference adjusts computational paths based on the complexity level of sign glosses, optimizing efficiency and accuracy. Experimental results demonstrate the effectiveness of Siformer.

**Strengths:**

1. Kinematic hand pose rectification: The authors propose a kinematic hand pose rectification approach that enforces constraints on hand joints based on their kinematic constraints. This approach enhances the realism of hand skeletal representations, resulting in more reasonable and lifelike representations of sign language glosses. The effectiveness of this approach is demonstrated through ablation studies, showing its contribution to the overall performance of Siformer.
2. Feature-isolated mechanism for robustness against missing data: The feature-isolated mechanism proposed in Siformer captures local spatial-temporal context concurrently and independently from individual features during the encoding phase. This approach decouples the interdependence between features and eases the impact of missing data on individual body parts during the recognition process. The authors demonstrate the robustness of Siformer against missing data through experiments with varying levels of occlusion on different body parts.
3. Input-adaptive inference for efficiency and accuracy: Siformer incorporates an input-adaptive inference mechanism that adjusts computational paths based on the complexity level of sign glosses. This mechanism utilizes internal classifiers at each layer to achieve early exiting when intermediate predictions remain unchanged for a predetermined number of consecutive times. The authors present a thorough analysis of the effectiveness of input-adaptive inference, demonstrating its ability to optimize both computational efficiency and accuracy with minimal additional effort in terms of model size and training time.

**Limitations:**

Lack of comparison with state-of-the-art methods on WLASL100: While the authors compare Siformer with previous state-of-the-art methods on the LSA64 dataset, they do not provide a comprehensive comparison on the WLASL100 dataset. The paper would benefit from a more extensive comparison with recent methods, such as SignBERT+ [13], to demonstrate the superiority of Siformer on this benchmark dataset.

Missing ablation study on the feature-isolated mechanism: Although the authors demonstrate the robustness of Siformer against missing data through experiments with varying levels of occlusion, they do not provide an ablation study specifically focusing on the contribution of the feature-isolated mechanism. An ablation study comparing Siformer with and without the feature-isolated mechanism would help to isolate its impact on the overall performance and robustness of the model.

Insufficient discussion on the choice of internal classifiers: The authors mention that they experimented with both trained internal classifiers and "brand-new" internal classifiers for input-adaptive inference, finding nearly identical results. However, they do not provide a detailed discussion on the choice of internal classifiers and the implications of using trained or "brand-new" classifiers. A more in-depth analysis of this aspect would provide valuable insights into the design choices of Siformer.

Lack of discussion on the potential limitations of the datasets: The paper does not discuss the potential limitations of the WLASL100 and LSA64 datasets, such as the diversity of signers, the range of sign glosses covered, and the quality of the skeletal data. Addressing these limitations and their potential impact on the generalizability of Siformer would strengthen the paper's contributions and provide a more comprehensive understanding of its performance in real-world scenarios.

**Suitability:**

2

---

### Official Review · Reviewer_Cqr4 · 2024-05-25

**Rating:** 5
**Confidence:** 3

**Summary:**

The paper titled "Siformer: Feature-isolated Transformer for Efficient Skeleton-based Sign Language Recognition" presents an advanced method for sign language recognition (SLR) utilizing a skeleton-based approach. The study highlights three main limitations in current skeleton-based SLR methods: lack of realistic hand pose representations, assumptions of complete data availability, and uniform treatment of sign glosses. To address these issues, the authors propose a kinematic hand pose rectification method, a feature-isolated mechanism for handling missing data, and an input-adaptive inference approach. The proposed method demonstrates state-of-the-art performance on WLASL100 and LSA64 datasets, achieving significant improvements in top-1 accuracy.

**Strengths:**

1.The paper introduces a kinematic hand pose rectification method to enhance the realism of hand skeletal representations, which is a significant advancement over existing methods.
2.The feature-isolated mechanism is a novel solution to the problem of missing data, focusing on capturing local spatial-temporal context independently, thereby improving the robustness of the SLR model.
3.The development of an input-adaptive inference approach to optimize computational efficiency and accuracy for varying complexity levels of sign glosses is a notable contribution.
4.The experimental results are strong, with the proposed method achieving a top-1 accuracy of 86.50% on WLASL100 and 99.84% on LSA64, representing substantial improvements over previous state-of-the-art methods.

**Limitations:**

1. The paper lacks a detailed explanation of the basis for selecting the alpha (α) value in the kinematic hand pose rectification method and how it affects model performance. A deeper theoretical basis or sensitivity analysis regarding α value selection would be beneficial.
2. There are potential errors in equations (7) and (8), particularly the presence of the V matrix within the softmax operation, which should be reviewed and corrected if necessary.
3. Figure 4 contains labeling issues, with two subfigures labeled as (a). This should be corrected to avoid confusion.

**Suitability:**

2

---

### Meta-Review · Area_Chair_dbyo · 2024-07-12

**Recommendation:** Accept (Poster)
**Confidence:** 3

**Metareview:**

The paper got two Borderline Accept (one downgraded from Weak Accept after the rebuttal) and 2 Borderline Reject (kept after the rebuttal) recommendations. Thus, the rebuttal didn’t have a positive effect on the reviewers’ ratings. Still, it was a borderline case discussed with other Area Chairs and Program Chairs (having an overall view of borderline cases), and the final recommendation is to accept this paper to be presented as a poster.